# Epileptiform Discharge and Electrographic Seizures during the Hypothermia Phase as Predictors of Rewarming Seizures in Children after Resuscitation

**DOI:** 10.3390/jcm9072151

**Published:** 2020-07-08

**Authors:** Jainn-Jim Lin, Mei-Hsin Hsu, Shao-Hsuan Hsia, Ying-Jui Lin, Huei-Shyong Wang, Hsuan-Chang Kuo, Ming-Chou Chiang, Oi-Wa Chan, En-Pei Lee, Kuang-Lin Lin

**Affiliations:** 1Division of Pediatric Critical Care and Pediatric Neurocritical Care Center, Chang Gung Children’s Hospital and Chang Gung Memorial Hospital, Chang Gung University College of Medicine, Taoyuan 333, Taiwan; lin0227@adm.cgmh.org.tw (J.-J.L.); tw1picu@gmail.com (S.-H.H.); ai3333@cgmh.org.tw (O.-W.C.); pilichrislnp@gmail.com (E.-P.L.); 2Graduate Institute of Clinical Medical Sciences, Chang Gung University, College of Medicine, Taoyuan 333, Taiwan; newborntw@gmail.com; 3Division of Pediatric Neurology, Chang Gung Children’s Hospital and Chang Gung Memorial Hospital, Chang Gung University College of Medicine, Taoyuan 333, Taiwan; wanghs444@cgmh.org.tw; 4Department of Respiratory Therapy, Chang Gung Children’s Hospital and Chang Gung Memorial Hospital, Chang Gung University College of Medicine, Taoyuan 333, Taiwan; 5Division of Neurology, Department of Pediatrics, Kaohsiung Chang Gung Memorial Hospital and Chang Gung University College of Medicine, Kaohsiung 833, Taiwan; a03peggy@cgmh.org.tw; 6Division of Critical Care, Department of Pediatrics, Kaohsiung Chang Gung Memorial Hospital and Chang Gung University College of Medicine, Kaohsiung 833, Taiwan; rayray@cgmh.org.tw (Y.-J.L.); kuohc117@adm.cgmh.org.tw (H.-C.K.); 7Division of Cardiology, Department of Pediatrics, Kaohsiung Chang Gung Memorial Hospital and Chang Gung University College of Medicine, Kaohsiung 833, Taiwan; 8Division of Neonatology, Chang Gung Children’s Hospital and Chang Gung Memorial Hospital, Chang Gung University College of Medicine, Taoyuan 333, Taiwan; 9Study Group for Intensive and Integrated Care of Pediatric Central Nervous System, Chang Gung Children’s Hospital, Taoyuan 333, Taiwan; bread86@cgmh.org.tw

**Keywords:** therapeutic hypothermia, predictors, rewarming seizure, children, resuscitation

## Abstract

The aim of this study was to determine the frequency, timing, and predictors of rewarming seizures in a cohort of children undergoing therapeutic hypothermia after resuscitation. We retrospectively reviewed consecutive pediatric patients undergoing therapeutic hypothermia after resuscitation admitted to our pediatric intensive care unit between January 2000 and December 2019. Continuous electroencephalographic monitoring was performed during hypothermia (24 h for cardiac aetiologies and 72 h for asphyxial aetiologies), rewarming (72 h), and then an additional 12 h of normothermia. Thirty comatose children undergoing therapeutic hypothermia after resuscitation were enrolled, of whom 10 (33.3%) had rewarming seizures. Two (20%) of these patients had their first seizure during the rewarming phase. Four (40%) patients had electroclinical seizures, and six (60%) had nonconvulsive seizures. The median time from starting rewarming to the onset of rewarming seizures was 37.3 h (range 6 to 65 h). The patients with interictal epileptiform activity and electrographic seizures during the hypothermia phase were more likely to have rewarming seizures compared to those without interictal epileptiform activity or electrographic seizures (*p* = 0.019 and 0.019, respectively). Therefore, in high-risk patients, continuous electroencephalographic monitoring for a longer duration may help to detect rewarming seizures and guide clinical management.

## 1. Introduction

Despite advances in the resuscitation of pediatric cardiac arrest, acute comatose status continues to be a challenging problem in pediatric post-cardiac arrest care [1]. Of the many aetiologies of acute coma, seizures, and especially nonconvulsive seizures, are common after resuscitation [2,3,4]. The frequency of seizures in children after resuscitation as assessed by continuous electroencephalographic (EEG) monitoring has been reported to range between 9.5% and 47% [5,6,7]. In addition, seizures after cardiac arrest are associated with poor outcomes [8], and prolonged seizures may contribute to the burden of secondary brain injury after cardiac arrest [9,10,11]. Therefore, the early identification of seizures may improve outcomes.

Therapeutic hypothermia has been demonstrated to have a neuroprotective effect against encephalopathy in animal models, perinatal asphyxia, and adult cardiac arrest [12,13,14]. One of the neuroprotective effects of therapeutic hypothermia is to reduce the release and accumulation of excitatory neurotransmitters during the ischemia and reperfusion phase after cardiac arrest and increase the threshold for seizures [12,13,14]. Conversely, fever can trigger and stimulate destructive processes [12]. Therefore, brain temperature may modulate epileptiform activity.

Few studies have investigated rebound seizures during the rewarming phase of therapeutic hypothermia in perinatal asphyxia and comatose children after cardiac arrest [5,15]. One proposed mechanism is a combination of increased neuronal excitability and excitatory neurotransmitter release, which may contribute to the development of rebound seizures seen during the rewarming phase of therapeutic hypothermia [16]. In addition, patients undergoing therapeutic hypothermia may require pharmacologic paralysis to manage shivering, and clinical seizures may be obscured by medications. Therefore, it is important to recognize that seizures may occur during the rewarming phase of therapeutic hypothermia. 

Although continuous EEG monitoring is recommended for comatose patients after cardiac arrest undergoing therapeutic hypothermia, continuous EEG monitoring is resource intensive and is not available at all institutions performing therapeutic hypothermia [17,18]. Identifying the frequency, timing, and predictors of rewarming seizures may help to optimize the use of continuous EEG monitoring. The aim of this study was to determine the frequency, timing, and predictors of rewarming seizures in a cohort of children undergoing therapeutic hypothermia after resuscitation.

## 2. Materials and Methods

### 2.1. Patient Population

This was a retrospective study using a chart review of patients who had successfully been resuscitated with recovery of spontaneous circulation (ROSC) and were undergoing therapeutic hypothermia at the pediatric intensive care unit of Chang Gung Children’s Hospital between 1 January 2010 and 31 December 2019 (Figure 1). Out-of-hospital cardiac arrest was defined as patients in whom chest compressions were initiated before arriving at the hospital, and in-hospital cardiac arrest was defined as patients in whom chest compressions were initiated in the emergency department or other hospital setting [19,20]. Asphyxial aetiologies were defined as resuscitation secondary to acute respiratory failure after evaluating all available data, and cardiac aetiologies were defined as resuscitation secondary to shockable rhythm, such as ventricular tachycardia or ventricular fibrillation [19,20,21,22]. Rewarming seizures were defined as seizures during the rewarming phase. 

The inclusion criteria were as follows: a: (1) age from 1 month to 18 years; (2) duration of cardiac arrest at least 3 minutes and ROSC after resuscitation; (3) comatose status (Glasgow Coma Scale ≤8) after ROSC; (4) received therapeutic hypothermia; and (5) received continuous EEG monitoring [19,20,21,22]. Patients were excluded if they met any of the following criteria: (1) age > 18 years; (2) did not receive continuous EEG monitoring or intermittent routine EEG; (3) not in a coma after resuscitation (Glasgow Coma Scale (GCS) > 8); and (4) hemodynamic instability refractory to intensive care and died within 72 h. All patients received therapeutic hypothermia within 6 h of resuscitation [19,20,21,22]. This study was approved by the Chang Gung Memorial Hospital Institutional Review Board (IRB numbers: 201600360B0D001 and 201900302B0).

### 2.2. Cooling Methods and Continuous EEG Monitoring Protocol

The cooling protocol for therapeutic hypothermia in the present study has been published previously [19,21]. In brief, therapeutic hypothermia to 33 °C was induced using external cooling pads (Arctic Sun, Medivance Inc. Louisville, CO, USA). The cooling duration was 24 h in children with cardiac aetiologies, and 72 h in children with asphyxial aetiologies. Then, the patients were slowly rewarmed by 1 °C per day until they reached 36 °C. All patients were sedated with midazolam and/or received neuromuscular blockers to prevent shivering during treatment. Sedation and neuromuscular blockers were stopped as soon as the body temperature was ≥36 °C [19,21].

EEG monitoring was performed using either a Stellate Harmonie (Natus; Pleasanton, CA, USA) or Nicolet Monitor (Natus Neuro, Middleton, USA) video-EEG system, and electrodes were placed according to the international 10–20 system. The patients were monitored with continuous video EEG monitoring, which was initiated as soon as possible after resuscitation and was performed during hypothermia (24 h for cardiac aetiologies and 72 h for asphyxial aetiologies), rewarming (72 h) and then for an additional 12 h of normothermia. During monitoring, EEG data were provided to the clinical teams at least twice a day, or more often if any change occurred including the occurrence of electrographic seizures. If the clinical team suspected that seizures may still occur, the monitoring was continued for longer. If electrographic seizures were identified, then monitoring was performed until at least 24 h after the end of the last electrographic seizure.

### 2.3. EEG Data Interpretation

EEG tracings were interpreted using standardized American Clinical Neurophysiology Society’s (ACNS) terminology [23]. The EEG background was scored as the most severe status and categorized during hypothermia as (1) normal (including sedated sleep), (2) slow–disorganized, (3) discontinuous or burst suppression, and (4) attenuated–featureless [6]. Any inter-ictal epileptiform discharges, including lateralized periodic discharges (previously termed as periodic lateralized epileptiform discharges), generalized periodic discharges (previously termed as generalized periodic epileptiform discharges), and isolated non-periodic spikes or sharp waves, were scored as present or absent [5]. Electrographic seizures were defined as an abnormal paroxysmal event that was different from the background, lasting >10 seconds (or shorter if associated with a clinical change) with a temporal–spatial evolution in morphology, frequency, and amplitude, and with a plausible electrographic field [11]. Electrographic seizures were classified as nonconvulsive seizures (no clinical signs observed by bedside caregivers or on video review) or electroclinical seizures (clinical abnormal stereotypic and paroxysmal movements associated with the EEG seizure) [11]. Nonconvulsive status epilepticus was defined according to the Salzburg consensus criteria for nonconvulsive status epilepticus [24].

### 2.4. Data Collection

The following information was collected from all patients: (1) demographics and pre-existing diseases; (2) event characteristics during cardiopulmonary resuscitation; (3) variables after resuscitation, including initial serum lactate level, post-cardiac arrest GCS; and (4) outcomes. Initial blood samples were collected within 1 hour of ROSC. The primary outcome was the presence or absence of rewarming seizures. The secondary outcomes were duration of hospitalization, 1-month mortality, and neurological outcomes, which were assessed using Pediatric Cerebral Performance Category (PCPC) scores in the children who survived for 6 months after the events or in those who died during follow-up [19,20,21,22,25]. PCPC scores measure the degree of cognitive function and are recorded as 1 to 6, where 1 is normal, 2 indicates mild disability, 3 indicates moderate disability, 4 indicates severe disability, 5 indicates coma or vegetative state, and 6 indicates brain death. Neurological outcomes were dichotomized as either a favorable prognosis (PCPC ≤ 2) or unfavorable prognosis (PCPC ≥ 3 or a change from baseline ≥ 1) [19,20,21,22,25]. In addition, time intervals from arrest to starting therapeutic hypothermia, from starting therapeutic hypothermia to continuous EEG monitoring placement, from arrest to continuous EEG monitoring placement, and from rewarming to the onset of rewarming seizures were also recorded.

### 2.5. Statistical Analysis

The patients were divided into rewarming seizures or non-rewarming seizures groups. Patient characteristics are presented as descriptive statistics, and the data are presented as mean± standard deviation. Between group differences were analyzed using the chi-squared test or Fisher’s exact test for categorical variables, and Student’s t-tests for normally distributed continuous variables. The Mann–Whitney U-test was used for non-normally distributed data.

To identify the factors associated with rewarming seizures, we performed logistic regression analysis with rewarming seizures as the dependent variable and demographic and EEG variables as the independent variables. In addition, associations between rewarming seizures and 1-month mortality and 6-month neurologic outcomes were also analyzed. Statistical analysis was performed using SPSS software, version 23.0 (IBM, Inc., Chicago, IL, USA). A two-sided p-value of < 0.05 was considered to indicate a statistically significant difference.

## 3. Results

### 3.1. Patients

During the 10-year study period, 62 children undergoing therapeutic hypothermia after resuscitation were identified, of whom 30 (48.4%) met the study entry criteria. Of the 32 children who were excluded, 15 received intermittent routine EEG, nine died within 72 h due to refractory cardiogenic shock despite the use of vasopressor and/or inotropic agents, and eight did not receive EEG examination. Twenty-four (80%) patients had out-of-hospital cardiac arrest, and six (20%) had in-hospital cardiac arrest. Thirteen (43.3%) of the 30 patients had a chronic illness before the cardiac arrest. Seven (53.8%) of 13 patients with chronic illness had neurologic illness before cardiac arrest, of which 3 had development delay, 2 had epilepsy, 1 had a brain tumor, and 1 had cerebral palsy. Twenty-nine (96.7%) had asphyxial etiologies, and only one (3.3%) had a cardiac etiology. The first documented arrest rhythm was described as asystole in 28 (93.3%) patients, bradycardia/pulseless electrical activity in one (3.3%) patient, and ventricular tachycardia/ventricular fibrillation in one (3.3%) patient. Nine (30%) of the 30 patients had acute seizures after resuscitation before therapeutic hypothermia. There were no significant differences in demographic data with respect to gender, age, the place of cardiac arrest, the presence of chronic illness, bystander-witnessed cardiac arrest, bystander-performed cardiopulmonary resuscitation, and the first documented arrest rhythm between the rewarming seizures and non-rewarming seizures groups. The demographic data are shown in Table 1.

### 3.2. Variables during and after Resuscitation

The mean duration of cardiac arrest to ROSC was 24.67 minutes (range 3 to 60 min). There were no significant differences in the duration between the rewarming seizures group (mean ± SE: 28.20 ± 15.55 min) and the non-rewarming seizures groups (mean ± SE: 22.90 ± 14.84 min, *p* = 0.342). Serum lactate and sugar levels immediately after resuscitation were similar between the rewarming seizures and non-rewarming seizures groups. The baseline patient characteristics including post-cardiac arrest GCS, admission Pediatric Logistic Organ Dysfunction Scores, and Pediatric Risk of Mortality were not significantly different between groups, suggesting a similar severity of illness after resuscitation. The event characteristics during resuscitation are listed in Table 2.

### 3.3. Survival Rate and Functional Outcomes

The overall 1-month mortality rate was 20%. There were no significant differences in 1-month mortality rate and duration of hospital stay between the rewarming seizures and non-rewarming seizures groups. Of the 24 survivors, only six (25%) had PCPC scores of 1 or 2 at 6 months of follow-up. The 6-month neurologic outcomes were better (PCPC ≤ 2) in the rewarming seizure group (3/8, 37.5%) than in the non-rewarming seizure group (3/13, 18.7%); however, this did not reach statistical significance (*p* = 0.362).

### 3.4. EEG Recording during Therapeutic Hypothermia and Rewarming Seizures

The mean time to starting therapeutic hypothermia after ROSC was 3.41 ± 1.71 h (range 0.5 to 6 h). The mean time to starting continuous EEG monitoring after therapeutic hypothermia was 5.72 ± 5.56 h (range 0 to 12 h), and the mean time between resuscitation from ROSC and continuous EEG recording was 7.43 ± 4.25 h (range 1 to 12 h). Table 3 shows the EEG findings and clinical details of the 30 comatose children after resuscitation.

Any epileptiform discharges (interictal, ictal, or both) during the hypothermia phase were present in 17 (56.7%) of the 30 patients. Interictal epileptiform discharges occurred during therapeutic hypothermia in 14 (46.7%) of the 30 patients, and they consisted of lateralized periodic discharges (6/14; 42.9%), generalized periodic discharges (6/14; 42.9%), and isolated non-periodic spikes or sharp waves (2/14, 14.2%). Electrographic seizures occurred during therapeutic hypothermia in 14 (46.7%) of the 30 patients, including five (35.7%) with electrographic status epilepticus. In addition, only five (35.7%) of the 14 patients with electrographic seizures had an eventual clear clinical correlate documented in their medical records. In terms of anticonvulsant treatment, 18 (60%) of 30 patients received anticonvulsant before the rewarming phase, including 8 (80%) of 10 patients in the rewarming seizures group and 10 (50%) of 20 patients in the non-rewarming group. Electrographic seizures were significantly associated with preceding interictal epileptiform activity (*p* = 0.003).

Since the rewarming protocol in our hospital is to slowly rewarm by 1 °C per day until the patient reaches 36 °C, the mean duration of EEG monitoring for therapeutic hypothermia was 163.56 ± 19.74 h. Ten (33.3%) of the 30 patients had rewarming seizures, of whom eight (80%) had rebound seizures during the rewarming phase, and two (20%) had a first seizure during the rewarming phase without prior seizure events (case 9 and 10). Four (40%) of the 10 patients had electroclinical seizures, and six (60%) had nonconvulsive seizures. Two (20%) of the 10 patients with rewarming seizures had electrographic status epilepticus. The time from rewarming to the onset of rewarming seizures was within 24 h in four (40%) patients, 24 to 48 h in two (20%) patients, and 48 to 72 h in four (40%) patients. The median time from rewarming to the onset of rewarming seizures was 37.3 h (range 6 to 65 h). No further seizures occurred during 12 h of normothermia in any of the patients. The patients with interictal epileptiform activity and electrographic seizures during the hypothermia phase were more likely to have rewarming seizures compared to those without interictal epileptiform activity or electrographic seizures (*p* = 0.019 and 0.019, respectively) (Table 4). In addition, rewarming seizures could not be predicted from any clinical or resuscitation variable.

## 4. Discussion

Identifying the frequency, timing, and predictors of rewarming seizures in comatose children undergoing therapeutic hypothermia after achieving ROSC may help to optimize the use of continuous EEG monitoring use. 

Rewarming during therapeutic hypothermia in pediatric post-cardiac arrest care is potentially more complex than the induction of cooling, because it can lead to destabilizing changes in neuronal activity [15]. Since therapeutic hypothermia has an anticonvulsant effect, seizures may develop during rewarming [14]. It remains unclear whether rewarming seizures further contribute to poor neurologic outcomes or are simply a marker of an irreversibly damaged brain [15]. In a pediatric study of continuous EEG monitoring after cardiac arrest, continuous EEG monitoring was performed during hypothermia (only 24 h), rewarming (only 12 to 24 h), and then an additional 24 h of normothermia. The authors found that the seizures started during the late hypothermic or rewarming periods (8/9), with the majority occurring during rewarming [5]. In our study, the mean duration of EEG monitoring for therapeutic hypothermia was 163.56 ± 19.74 h. Ten (33.3%) of our 30 patients had rewarming seizures, of whom six (60%) had nonconvulsive seizures and two (20%) had electrographic status epilepticus. The median time from rewarming to the onset of electrographic seizures was 37.3 h (range 6 to 65 h). Therefore, continuing EEG monitoring during rewarming until normothermia is achieved in patients undergoing therapeutic hypothermia is recommended.

As a result of the high incidence of electrographic seizures after ROSC, continuous EEG monitoring has increasingly been used in pediatric post-cardiac arrest care [26,27]. Furthermore, patients undergoing therapeutic hypothermia may require pharmacologic paralysis to manage shivering, and clinical seizures may be obscured by medications. Therefore, the accurate identification of nonconvulsive seizures is important, and continuous EEG monitoring is required to detect electrographic seizure detection. A recent consensus statement from the ACNS Guidelines recommends continuous EEG monitoring for pediatric patients who remain comatose after cardiac arrest to identify electrographic seizures [17,18]. In addition, the statement recommends initiating EEG monitoring as soon as possible, continuing monitoring for 24 to 48 h in most patients, but continuing until after 24 h of normothermia in patients undergoing therapeutic hypothermia [17,18]. 

However, fewer studies have focused on the predictors of rewarming seizures in children undergoing therapeutic hypothermia after resuscitation [5]. A pediatric study investigating the timing of seizure development during hypothermia after cardiac arrest using continuous EEG monitoring reported that most seizure activity started prior to the rewarming phase. The authors concluded that seizures occurred more often in patients with more abnormal EEG background patterns (e.g., excessive discontinuity, burst suppression, or highly attenuated featureless tracings) [5]. In our study, the patients with interictal epileptiform activity and electrographic seizures during the hypothermia phase were more likely to have rewarming seizures compared to those without interictal epileptiform activity or electrographic seizures (*p* = 0.019 and 0.019, respectively). In addition, seizures could not be predicted from any clinical or resuscitation variable. Therefore, identifying predictors of rewarming seizures may help to optimize the use of continuous EEG monitoring, especially in institutions with limited resources.

There are several limitations to this study. First, this is a small, retrospective single-center study. A larger, multi-center study using common data definitions could provide more precise estimates of the frequency of rewarming seizures in this population. Second, our protocol routinely treats comatose patients undergoing therapeutic hypothermia after resuscitation with benzodiazepines during the induction, maintenance, and rewarming phases of therapeutic hypothermia, and benzodiazepines may alter seizure frequency and may increase the frequency of burst-suppression patterns. In addition, hypothermia may also decrease brain activity. These factors may explain the high rate of abnormal EEG background, such as burst-suppression and low flat patterns in this study. Future studies should consider more frequent interruptions of sedation for EEG interpretation. Third, prospective studies are needed to determine the impact of rewarming seizures on patient outcomes, and whether treatment for rewarming seizures improves patient outcomes. Finally, additional studies are needed to determine whether an altered rewarming regime would improve rewarming seizures after therapeutic hypothermia.

## 5. Conclusions

Our data may be clinically helpful as predictors of rewarming seizures in children undergoing therapeutic hypothermia after resuscitation. The patients who had interictal epileptiform activity or electrographic seizures during the hypothermia phase had a risk of rewarming seizures. Therefore, in high-risk groups, continuous EEG monitoring for a longer duration may help to detect rewarming seizures and guide clinical management. These data may also help to optimize the use of EEG monitoring in institutions with limited resources. 

## Figures and Tables

**Figure 1 jcm-09-02151-f001:**
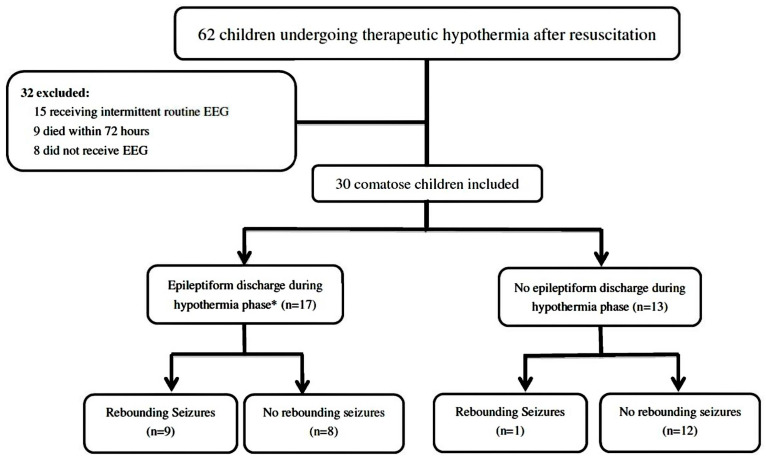
During the 10-year study period, 62 children undergoing therapeutic hypothermia after resuscitation were identified, of whom 30 (48.4%) patients met the study entry criteria. We excluded 32 children, including 15 who received intermittent routine EEG, nine who died within 72 h due to refractory cardiogenic shock despite the use of vasopressor and/or inotropic agents, and eight who did not receive EEG monitoring. Epileptiform discharges (interictal, ictal, or both) during the hypothermia phase were noted in 17 (56.7%) patients, and 10 (33.3%) patients had rewarming seizures. (EEG: electroencephalography). * Inter-ictal epileptiform discharges, including lateralized periodic discharges, generalized periodic discharges, and isolated non-periodic spikes or sharp waves.

**Table 1 jcm-09-02151-t001:** The characteristics of 30 children undergoing therapeutic hypothermia after resuscitation.

Characteristic	Rewarming Seizures (*n* = 10)	Non-Rewarming Seizures (*n* = 20)	*p* Value
Gender			1.000
Female	3 (30%)	5 (25%)	
Male	7 (70%)	15 (75%)	
Age			0.842
1–11 months	3 (30%)	8 (45%)	
1–4 years	3 (30%)	7 (30%)	
5–8 years	2 (20%)	3 (15%)	
9–18 years	2 (20%)	2 (10%)	
Chronic pre-existing illness			0.992
No	6 (60%)	11 (55%)	
Respiratory	1 (10%)	2 (10%)	
Neurologic	2 (20%)	5 (25%)	
Other	1 (10%)	2 (10%)	
Cardiac arrest location			0.372
OHCA	7 (70%)	17 (85%)	
IHCA	3 (30%)	3 (15%)	
Bystander-witnessed cardiac arrest	4 (40%)	13 (65%)	0.255
Bystander-performed CPR	3 (30%)	12 (60%)	0.245
Initial rhythm			0.284
Asystole	9 (90%)	19 (95%)	
Bradycardia/PEA	0	1 (5%)	
VT/Vf	1 (10%)	0	
Seizures before therapeutic hypothermia	2 (20%)	7 (35%)	0.675

OHCA: out-of-hospital cardiac arrest; IHCA: in-hospital cardiac arrest; CPR: cardiopulmonary resuscitation; PEA: pulseless electrical activity; VT: ventricular tachycardia; Vf: ventricular fibrillation.

**Table 2 jcm-09-02151-t002:** Variables during and after resuscitation and outcomes of the 30 children undergoing therapeutic hypothermia after resuscitation.

Characteristics	Rewarming Seizures (*n* = 10)	Non-Rewarming Seizures (*n* = 20)	*p* Value
Characteristics during resuscitation			
Interval from CPR to ROSC (min)	28.20 ± 15.55	22.90 ± 14.84	0.342
Serum pH	7.21 ± 0.18	7.10 ± 0.21	0.171
Initial glucose (mg/dL)	293.60 ± 128.64	334.00 ± 204.24	0.576
Initial lactate (mmol/L)	83.66 ± 41.53	87.29 ± 42.37	0.840
Post-cardiac arrest GCS	3.70 ± 1.15	3.70 ± 1.55	1.000
PRISM score	34.70 ± 5.61	38.60 ± 8.89	0.218
PLODS	38.60 ± 11.77	35.95 ± 8.39	0.536
Outcome			
Hospital length of stay (days)	53.80 ± 30.75	50.20 ± 39.94	0.805
1-month mortality	2 (20%)	4 (20%)	1.000
6-month neurologic outcome (*n* = 24)			0.362
Good prognosis (PCPC score ≤ 2)	3 (37.5%)	3 (18.7%)	
Poor prognosis (PCPC score ≥ 3)	5 (62.5%)	13 (81.3%)	

CPR: cardiopulmonary resuscitation; ROSC: return of spontaneous circulation; GCS: Glasgow Coma Scale; PRISM: pediatric risk of mortality; PLODS: pediatric logistic organ dysfunction scores; PCPC: pediatric cerebral performance category.

**Table 3 jcm-09-02151-t003:** EEG findings and clinical details of the 30 comatose children after resuscitation.

Age (y)/Sex/Cardiac Arrest Location		Seizure before TH	Hypothermia Phase			Rewarming Phase			LOS (Days)	6-Month PCPC
			EEG BG ^#^	ED	ES	EEG BG ^#^	ES	Seizure Onset (h) *		
**Rewarming Seizures Group**										
**Any ED or ES during the Hypothermia Phase**										
1	6.49/M/OHCA	Y	3	GED	CSE	3	CSE	6	105	3
2	3.17/M/OHCA	Y	3	N	CSE	3	NCS	65	27	-^a^
3	4.33/M/OHCA	N	3	GED	CSE	3	CSE	37	50	3
4	4.39/M/IHCA	N	3	GED	NCS	3	NCS	14	48	1
5	0.80/M/OHCA	N	3	PLED	NCS	3	NCS	62	47	3
6	0.21/M/IHCA	N	3	FED	NCS	3	NCS	22	50	3
7	7.22/M/OHCA	N	2	PLED	NCS	2	NCS	15	18	-^a^
8	0.53/F/OHCA	N	2	PLED	NCS	3	NCS	53.5	66	3
9	16.52/F/OHCA	N	2	FED	N	2	NCS	46.5	22	1
**No ED or ES during the Hypothermia Phase**										
10	13.04/F/IHCA	N	3	N	N	3	NCS	52	105	2
**Non-rewarming seizures group**										
**Any ED or ES during the Hypothermia Phase**										
11	3.48/M/OHCA	Y	3	GED	N	4	N	-	11	-^a^
12	7.26/M/OHCA	Y	3	N	NCSE	4	N	-	12	-^a^
13	3.78/M/OHCA	Y	3	N	ECS	4	N	-	9	-^a^
14	1.16/M/OHCA	Y	2	PLED	NCS	2	N	-	37	3
15	10.75/F/IHCA	Y	2	PLED	ECS	2	N	-	39	3
16	4.11/F/OHCA	N	3	GED	NCSE	3	N	-	110	4
17	1.79/M/OHCA	N	3	GED	NCS	3	N	-	24	2
18	0.47/M/OHCA	N	2	PLED	N	3	N	-	72	5
**No ED or ES during the Hypothermia Phase**										
19	0.02/F/OHCA	Y	3	N	N	4	N	-	34	3
20	1.04/M/OHCA	Y	2	N	N	2	N	-	14	1
21	0.45/M/OHCA	Y	2	N	N	2	N	-	28	3
22	0.16/M/OHCA	N	4	N	N	4	N	-	67	3
23	0.54/M/OHCA	N	4	N	N	4	N	-	138	4
24	0.41/M/OHCA	N	4	N	N	4	N	-	43	3
25	0.18/M/OHCA	N	4	N	N	4	N	-	42	5
26	0.23/M/OHCA	N	4	N	N	4	N	-	23	-^a^
27	4.59/F/IHCA	N	3	N	N	3	N	-	68	5
28	15.4/M/OHCA	N	3	N	N	3	N	-	53	3
29	5.87/M/OHCA	N	2	N	N	2	N	-	41	3
30	8.47/F/IHCA	N	1	N	N	1	N	-	15	1

y: years; F: female; M: male; IHCA: in-hospital cardiac arrest; OHCA: out-of-hospital cardiac arrest; TH: therapeutic hypothermia; h: hours; Y: yes; N: No; EEG: electroencephalography; BG: background; ED: epileptiform discharge; ES: electrographic seizures; ECS: electroclinical seizures; CSE: convulsive status epilepticus; NCS: nonconvulsive seizure; NCSE: nonconvulsive status epilepticus; PLEDs: periodic lateralized epileptiform discharges; GPEDs: generalized periodic epileptiform discharges; FED: focal epileptiform discharge; LOS: hospital length of stay; PCPC: Paediatric Cerebral Performance Category;. ^#^ EEG Background was categorized as (1) normal (including sedated sleep), (2) slow–disorganized, (3) discontinuous or burst suppression, and (4) attenuated–featureless. * time intervals from rewarming to onset of rewarming seizures. -^a^ Died within 1 month.

**Table 4 jcm-09-02151-t004:** EEG finding during hypothermia phase in the 30 children after resuscitation.

Characteristics	Rewarming Seizures (*n* = 10)	Non-Rewarming Seizures (*n* = 20)	Odds Ratio	95% CI	*p* Value
**Duration of EEG Monitoring (hours)**	181.17 ± 17.33	153.66 ± 13.15	-	-	<0.001 *
**Background during the Hypothermia Phase**			0.76	0.275–2.125	0.606
Normal	0	1 (5%)			
Slow–disorganized	3 (30%)	6 (30%)			
Discontinuous or burst suppression	7 (70%)	8 (40%)			
Attenuated–featureless	0	5 (25%)			
**Epileptiform Discharge (Interictal, Ictal or both) during the Hypothermia Phase**			13.50	1.42–128.25	0.023 *
Yes	9 (90%)	8 (45%)			
No	1 (10%)	12 (55%)			
**Epileptiform Discharge (Interictal)**			9.33	1.51–57.65	0.016 *
Yes	8 (80%)	6 (40%)			
No	2 (20%)	14 (60%)			
**Electrographic Seizures**			9.33	1.51–57.65	0.016 *
Yes	8 (80%)	6 (20%)			
No	2 (30%)	14 (80%)			

EEG: electroencephalography; CI: confidence interval. * statistically significant: *p* < 0.05.

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
