# Peer review of "Epileptiform Discharge and Electrographic Seizures during the Hypothermia Phase as Predictors of Rewarming Seizures in Children after Resuscitation"

_jcm, 2020, doi:10.3390/jcm9072151_

Round 1

Reviewer 1 Report

Lin et al. investigate the frequency, timing and predictors of rewarming seizures in a cohort of children undergoing therapeutic hypothermia after resuscitation. They found that patients with epileptiform discharges during hypothermia have an higher risk of rewarming seizures.

I congratulate with the Authors because the study is interesting and technically well designed with a robust protocol. In particular, electroencephalographic findings are very well described.

I suggest few minor changes:

Page 1, lines 44-45. This sentence is a repetition of the previous sentence and, therefore, it should be deleted.

Page 2, line 84. Delete “cardiac arrest”

Page 6, line 212. Delete “significantly”. This comparison was not statistically significant.

The Authors state that thirteen patients have chronic illness. Anyone of those patients suffered from epilepsy or other neurological condition that could increase the risk of seizures? This point should be clarified in the text.

I suggest deleting lines 259-264 (discussion section) because what Authors state is already clearly exposed in the "Results" section.

Author Response

Dear professor:

Re: Manuscript ID: jcm-811072
Title: Epileptiform Discharge and Electrographic Seizures during the Hypothermia Phase as Predictors of Rewarming Seizures in Children after Resuscitation
Authors: Jainn-Jim Lin, Mei-Hsin Hsu, Shao-Hsuan Hsia, Ying-Jui Lin, 
Huei-Shyong Wang, Hsuan-Chang Kuo, Ming-Chou Chiang, Oi-Wa Chan, En-Pei Lee, Kuang-Lin Lin *

Please find the revised version of manuscript for consideration of publication in the Journal of Clinical Medicine. My coauthors have all read and agreed with the revision what we have tried our best according to your comments. The changes that have been made were described in detail as follows:

Response to Reviewer 1 Comments

Lin et al. investigate the frequency, timing and predictors of rewarming seizures in a cohort of children undergoing therapeutic hypothermia after resuscitation. They found that patients with epileptiform discharges during hypothermia have an higher risk of rewarming seizures.

I congratulate with the Authors because the study is interesting and technically well designed with a robust protocol. In particular, electroencephalographic findings are very well described.

I suggest few minor changes:

Point 1: Page 1, lines 44-45. This sentence is a repetition of the previous sentence and, therefore, it should be deleted.

Response 1: Thanks for your comments. According to your comment, we have deleted the sentence in the page 1 line 47 to 48 as following: “Patients with interictal epileptiform activity or electrographic seizures during the hypothermia phase had a risk of rewarming seizures.

Point 2: Page 2, line 84. Delete “cardiac arrest”

Response 2: Thanks for your comments. According to your comment, we have deleted the phrase in the page 2 line 87 as following: “Out-of-hospital cardiac arrest cardiac arrest was defined as patients in whom chest compressions were initiated before arriving at the hospital,…”

Point 3: Page 6, line 212. Delete “significantly”. This comparison was not statistically significant.

Response 3: Thanks for your comments. According to your comment, we have deleted the word in the page 6 line 218 as following: “The 6-month neurologic outcomes were significantly better (PCPC ≤ 2) in the rewarming seizure group (3/8, 37.5%) than in the non-rewarming seizure group (3/13, 18.7%); however, this did not reach statistical significance (p = 0.362).”

Point 4: The Authors state that thirteen patients have chronic illness. Anyone of those patients suffered from epilepsy or other neurological condition that could increase the risk of seizures? This point should be clarified in the text.

Response 4: Thanks for your comments. 7 (53.8%) of 13 patients with chronic illness had neurologic illness before cardiac arrest, including 3 had development delay, 2 had epilepsy, 1 had brain tumor and 1 had cerebral palsy. And we added above description in the result 3.1 patient in the page 5 line 184 to 185.

Point 5: I suggest deleting lines 259-264 (discussion section) because what Authors state is already clearly exposed in the "Results" section.

Response 5: Thanks for your comments. According to your comment, we have deleted the sentence in the page 14 line 271 to 276 as following: “In this study, 10 (33.3%) of the 30 patients had rewarming seizures. Eight (80%) of these 10 patients had rebound seizures during the rewarming phase, and two (20%) had a first seizure during the rewarming phase without prior seizure events. The patients with interictal epileptiform activity and electrographic seizures during the hypothermia phase were more likely to have rewarming seizures compared to those without interictal epileptiform activity or electrographic seizures (p = 0.019 and 0.019, respectively).

Sincerely yours,

Kuang-Lin Lin, MD

Division of Pediatric Neurology, Chang Gung Children’s Hospital, 5 Fu-Shin Street, Kwei-Shan, Taoyuan, 333, Taiwan

Tel: +886-3-3281200 ext. 8200

Fax: +886-3-3288957

Reviewer 2 Report

This is a retrospective study that analyzed the rewarming seizures in a cohort of children who underwent therapeutic hypothermia after resuscitation. This topic is very interesting and the results should have clinical significance on the treatment of these children. However, there are some concerns pertaining to the study rationale, sample size and statistical inference, which limits the accuracy of results and the enthusiasm on the conclusion.

  1. In abstract, the authors state “We hypothesized that a substantial number of children undergoing therapeutic hypothermia after resuscitation may have rewarming seizures”. To test this hypothesis, two groups of objects (patients) should be included: one group of children who underwent therapeutic hypothermia after resuscitation and the other group of children who did not undergo any therapeutic hypothermia. Initial point of comparison would be the percentage of how many patients had seizures after resuscitation. If there is a difference between these two groups, further analysis can be conducted. While in this study, the authors only included children who underwent therapeutic hypothermia after resuscitation and analyzed the seizures in these patients. These data are meaningful, but are not sufficient to either accept or refuse the hypothesis.
  2. This is a retrospective study which requires a relatively large sample size. The authors have 30 children included in this study. Looks like the authors are from multiple hospitals, is it possible to increase the sample size?

Author Response

Dear professor:

Re: Manuscript ID: jcm-811072
Title: Epileptiform Discharge and Electrographic Seizures during the Hypothermia Phase as Predictors of Rewarming Seizures in Children after Resuscitation
Authors: Jainn-Jim Lin, Mei-Hsin Hsu, Shao-Hsuan Hsia, Ying-Jui Lin, 
Huei-Shyong Wang, Hsuan-Chang Kuo, Ming-Chou Chiang, Oi-Wa Chan, En-Pei Lee, Kuang-Lin Lin *

Please find the revised version of manuscript for consideration of publication in the Journal of Clinical Medicine. My coauthors have all read and agreed with the revision what we have tried our best according to your comments. The changes that have been made were described in detail as follows:

Response to Reviewer 2 Comments

This is a retrospective study that analyzed the rewarming seizures in a cohort of children who underwent therapeutic hypothermia after resuscitation. This topic is very interesting and the results should have clinical significance on the treatment of these children. However, there are some concerns pertaining to the study rationale, sample size and statistical inference, which limits the accuracy of results and the enthusiasm on the conclusion.

Point 1: 1. In abstract, the authors state “We hypothesized that a substantial number of children undergoing therapeutic hypothermia after resuscitation may have rewarming seizures”. To test this hypothesis, two groups of objects (patients) should be included: one group of children who underwent therapeutic hypothermia after resuscitation and the other group of children who did not undergo any therapeutic hypothermia. Initial point of comparison would be the percentage of how many patients had seizures after resuscitation. If there is a difference between these two groups, further analysis can be conducted. While in this study, the authors only included children who underwent therapeutic hypothermia after resuscitation and analyzed the seizures in these patients. These data are meaningful, but are not sufficient to either accept or refuse the hypothesis.

Response 1: Thanks for your comments. As your comments, it is important to compare the frequency of seizures between therapeutic hypothermia and non-therapeutic hypothermia. I think that will be the study to compare the effect of therapeutic hypothermia on seizures. But in our study, we focus on the frequency of seizure during the rewarming phase in children who underwent therapeutic hypothermia after resuscitation. Therefore, we revised the sentence as following in the abstract in the page 1 line 32 to 33 “We hypothesized that a substantial number of children undergoing therapeutic hypothermia after resuscitation may have seizures during rewarming phase”.

Point 2: 2. This is a retrospective study which requires a relatively large sample size. The authors have 30 children included in this study. Looks like the authors are from multiple hospitals, is it possible to increase the sample size?

Response 2: Thanks for your comments. All the authors were from 2 hospital of the Chang Gung Children’s hospital, Linkou and Kaohsiung branch. Although continuous EEG monitoring is recommended for comatose patients after cardiac arrest undergoing therapeutic hypothermia, we only have one continuous EEG monitoring machine in Linkou branch since 2007 and one in Kaohsiung branch since 2018. Therefore, it is difficult to perform continuous EEG monitoring for a long time in all patients. In our study, during the 10-year study period, 62 children undergoing therapeutic hypothermia after resuscitation were identified, of whom 30 (48.4%) met the study entry criteria. Of the 32 children who were excluded, 15 received intermittent routine EEG, nine died within 72 hours due to refractory cardiogenic shock despite the use of vasopressor and/or inotropic agents, and eight did not receive EEG examination. Therefore, it may be difficult to increase number at this moment. But it could be increased cases in the future.

Sincerely yours,

Kuang-Lin Lin, MD

Division of Pediatric Neurology, Chang Gung Children’s Hospital, 5 Fu-Shin Street, Kwei-Shan, Taoyuan, 333, Taiwan

Tel: +886-3-3281200 ext. 8200

Fax: +886-3-3288957

Round 2

Reviewer 2 Report

Thank the authors for the clarification.

I still don't think the revised hypothesis can be either accept or refused by the data. Based on the clarification, I suggest to delete that sentence to avoid confusing. The following sentence "The aim of this study ..." described it more accurate.

Author Response

Response to Reviewer 2 Comments

Point 1: Thank the authors for the clarification.

I still don't think the revised hypothesis can be either accept or refused by the data. Based on the clarification, I suggest to delete that sentence to avoid confusing. The following sentence "The aim of this study ..." described it more accurate.

Response 1: Thanks for your comments. As your comments, we deleted the following sentence in the abstract in the page 1 line 32 to 33 “We hypothesized that a substantial number of children undergoing therapeutic hypothermia after resuscitation may have seizures during rewarming phase.”

Sincerely yours,

Kuang-Lin Lin, MD

Division of Pediatric Neurology, Chang Gung Children’s Hospital, 5 Fu-Shin Street, Kwei-Shan, Taoyuan, 333, Taiwan

Tel: +886-3-3281200 ext. 8200

Fax: +886-3-3288957
